# DEIT-Based Bone Position and Orientation Estimation for Robotic Support in Total Knee Arthroplasty—A Computational Feasibility Study

**DOI:** 10.3390/s24165269

**Published:** 2024-08-14

**Authors:** Jakob Schrott, Sabrina Affortunati, Christian Stadler, Christoph Hintermüller

**Affiliations:** 1Institute of Measurement Technology, Johannes Kepler University, 4020 Linz, Austria; 2Department for Orthopedics and Traumatology, Kepler University Hospital, 4020 Linz, Austria; 3Medical Faculty, Johannes Kepler University, 4020 Linz, Austria; 4Institute for Biomedical Mechatronics, Johannes Kepler University, 4020 Linz, Austria

**Keywords:** differential electrical impedance tomography, position estimation, axial orientation estimation, total knee arthroplasty, robotic support

## Abstract

Total knee arthroplasty (TKA) is a well-established and successful treatment option for patients with end-stage osteoarthritis of the knee, providing high patient satisfaction. Robotic systems have been widely adopted to perform TKA in orthopaedic centres. The exact spatial positions of the femur and tibia are usually determined through pinned trackers, providing the surgeon with an exact illustration of the axis of the lower limb. The drilling of holes required for mounting the trackers creates weak spots, causing adverse events such as bone fracture. In the presented computational feasibility study, time differential electrical impedance tomography is used to locate the femur positions, thereby the difference in conductivity distribution between two distinct states $s_{0}$ and $s_{1}$ of the measured object is reconstructed. The overall approach was tested by simulating five different configurations of thigh shape and considered tissue conductivity distributions. For the cylinder models used for verification and reference, the reconstructed position deviated by about ≈1 mm from the actual bone centre. In case of models mimicking a realistic cross section of the femur position deviated between 7.9 mm 24.8 mm. For all models, the bone axis was off by about φ=1.50° from its actual position.

## 1. Introduction

Total knee arthroplasty (TKA) is a well-established and successful treatment option for patients with end-stage osteoarthritis of the knee. It provides high patient satisfaction with significantly improved quality of life [1]. In recent years, robotic systems have been widely adopted for and are frequently used to perform TKA in orthopaedic centres [2]. The precise measurements performed by robotic systems and the virtual illustration of the patient’s anatomy help to minimize potential errors caused by inaccurate human estimations regarding the position and axes orientation of femur and tibia [3,4,5,6,7]. These errors directly affect the alignment of saw jigs certain angles relative to the presumed axis of the femur when performed by the surgeon manually. These axes are crucial when performing TKA. They significantly influence the composition and bio-mechanics of the knee joint, the ligament tension, motion sequence, and the location of cartilaginous defects in osteoarthritic conditions [8,9].

Dependent on the chosen TKA alignment philosophy, the tibial and femoral component of the implant are placed in specific angles relative to the axes of the femur and tibia during TKA [10,11]. For example, when aiming at mechanical alignment, the components are place in 90° to the mechanical axis of each bone. In kinematic alignment, in contrast, the goal is to restore the patient’s pre-arthritic joint line.

The configuration of the patient’s leg, the axes of femur and tibia, are preoperatively determined from an X-ray image or a computed tomography (CT) scan of the entire leg where the patient is exposed to a relevant amount of ionizing radiation [7,12]. Intraoperatively, many robotic systems use pinned trackers screwed directly into the femur and tibia, to determine the exact spatial position of these two bones. These are used to present the surgeon with an exact illustration of the axis of the lower limb [13,14]. While this method provides a high grade of precision, the invasiveness of drilling holes into the patient’s femur and tibia comes with several relevant disadvantages and possible adverse events. These include wound healing disorders or fractures caused by the weak spots created by the drill holes required for mounting the spatial trackers [15,16,17].

Therefore, a non-invasive system able to accurately determine the axes of femur and tibia could provide significant benefits in (i) reducing the patient’s radiation exposure and (ii) minimizing the risk of adverse events associated with the placement of the pins required for proper estimation of bone position and orientation in robotic-assisted TKA.

Electrical impedance tomography (EIT) is a non-invasive method which allows to capture the location and distribution of tissues inside the body [18,19]. The EIT images are generated by iteratively applying small currents ≤10 mA to the investigated object for example the thigh or lower limb and measuring the resulting potential difference observable at the measurement electrodes [20]. Figure 1 shows the envisioned configuration of an EIT system used to compute as accurate as possible estimates for the axis position and orientation of the femur and the tibia. The electrodes indicated by the red circles in Figure 1a are used to inject the currents and the blue ones are used to measure the voltage distribution resulting from the corresponding electrical field. The black electrodes symbolize the common ground electrode used for the current injection and measuring the voltage differences.

The advantage of EIT is that it does not require any ionizing radiation compared to the X-ray or CT imaging methods utilized by the aforementioned marker-based approaches. Apart from standard applications like functional lung imaging and diagnosis of lung diseases [21,22,23,24] EIT is used to assess cardiovascular function [25,26], monitor blood flow in the brain perfusion [27,28], cancer diagnosis and treatment [29,30] as well as monitoring tissue water content [31]. It has been used to generate cross section images of the forearm [32,33], analyse the structure of trabecular bone [34,35], and assess degenerative processes for example related to osteoarthritis [36]. In recent studies, it is used to monitor muscle engagement in unsupervised rehabilitation training [37] as well as personal fitness [38].

Similar to the EIT-based navigation tool for total hip surgery developed by Ren et al. [39,40] and prosthesis and bone fracture monitoring developed by [41] it is the aim of the presented computational feasibility study to test whether standard differential EIT methods can be used to locate the central axis of the femur inside the thigh. In addition to the axis position also the ability to provide an accurate estimate of the axial orientation is tested. The obtained results are used to obtain knowledge about the suitability and feasibility of the chosen approach and which changes and improvements will be necessary to obtain accurate positional estimates within positional variations of less than a few millimeters and deviations of it orientation of ≤2°. The governing idea, envisioned in Figure 1, is to reduce TKA-related wound healing disorders and avoid TKA induced weak spots in the bone structure by replacing the marker-based approaches to determine the axes position and orientation of tibia and femur by simple EIT-based measurements on the lower limbs.

Section 2 briefly outlines the standard DEIT method used compute the 3D conductivity distribution and the basic 3D segmentation algorithm for locating the central bone axis and estimating its orientation. The obtained results are presented in Section 3 and discussed in Section 4. Section 4 analyses the obtained results and provides some considerations on necessary changes, adoptions and further improvements to obtain sufficiently accurate position and orientation estimates. These are crucial for the envisioned approach and for reliably distinguishing the tibia from the fibula which is not part of the knee joint.

## 2. Materials and Methods


There exist two basic EIT principles. The absolute EIT (AEIT) computes the conductivity distribution *a* causing the measured voltage distributions *V* as a response to the injected current sequences *I* [20,42]. The results obtained for *a* are highly sensitive to errors in the geometric model of the imaged volume and deviations between actual and modelled electrode position [43].

In this computational study, the time differential EIT (DEIT/TDEIT) approach [20,44,45] is used to demonstrate that EIT can be used locate the femur within the thigh. Instead of reconstructing the distribution of *a* directly, the relative changes in *a* between two time points $t_{0}$ and $t_{1}$ or two distinct states $s_{0}$ and $s_{1}$ of the measured object are reconstructed. At both time points $t_{0}$ and $t_{1}$ for both states $s_{0}$ and $s_{1}$, respectively, EIT measurements are performed. Each of the two distinct recordings consists of several individual pairs of applied current patterns $I_{0}$, $I_{1}$ and corresponding voltage measurements $V_{0}$, $V_{1}$ resulting from differences Δa in the conductivity distribution *a* within the measured object observable over time. This approach is, for example, used in functional lung diagnostics to estimate the breathing volume [21,22]. By those means, $I_{0}$, $V_{0}$ are recorded at the end of expiration phase and $I_{1}$, $V_{1}$ during inspiration phase.

Section 2.1 provides a brief outline of the chosen DEIT approach, which is used in the presented feasibility study to compute Δa between the actual $a_{1}$ tissue conductivity distribution of the subjects thigh and a digital model where the femur has been removed. This is achieved by replacing the conductivities of the bone $a_{bone}$ and bone marrow $a_{marrow}$, depicted in Figure 1b in blue, within the modelled conductivity distribution $a_{0}$ by the conductivity of muscle $a_{muscle}$. The resulting finite element model (FEM) of the thigh is used to compute an estimate of $V_{0}$. The FEM can be created from any set of images representing the 3D structure of the thigh or lower leg, e.g., X-ray or CT. In the present study, it is generated using the netgen library interfaces provided by the EIDORS [46] Matlab library (Version 3.10).

The realistic model includes compartments for the skin, the subcutaneous fat, the muscle and the intramuscular fat. The simplest one is just a cylinder with a constant value for $a_{0}$. These models are used to simulate the electrode potential distribution $V_{0}$ for a boneless thigh resulting from a set of predefined current patterns $I_{p}$ applied to the thigh. In the next step, an estimate of Δa is reconstructed using the EIDORS Matlab library [46]. In the last two steps, the 3D FEM model is segmented into bone structure and background representing any other tissue based upon Δa. The centre points $p_{c}$ of the FEM elements classified as tibial bone structure are used to estimate its position $p_{b}$ in the plane defined by the electrodes and the orientation of its central axis $n_{b}$.

### 2.1. Forward Simulation and Inverse Estimation of Conductivity

The $V_{0}$ values are obtained by solving the EIT forward problem [20,47] for a given set of current patterns *I* and the assumed conductivity distribution $D(a_{0})$ of a thigh, where the conductivity $a_{bone}$ and $a_{marrow}$ for the cortical bone and bone marrow have been replaced by the conductivity $a_{muscle}$ of the skeletal muscle.
(1)$$V_{0}=\frac{I_{0}}{Y(a)}\;\mathrm{with}:\;Y\left(a\right)=C^{T}S\left(r\right)D\left(a_{0}\right)C$$
The matrices C represent the connectivity matrix of the FEM model describing the individual elements and the system matrix S(r) describing its overall geometric properties of the FEM model. Computations are performed using the EIDORS Matlab library [46], which is also used to solve the DEIT inverse problem to reconstruct Δa from the voltage difference $\Delta V=V_{1}-V_{0}$.
(2)$$\varepsilon ={∥\Delta V-F(\Delta a)∥}_{W}^{2}+\lambda^{2}{∥a-a_{0}∥}_{Q}^{2}=min\;F\left(\Delta a\right)=J\Delta a$$
The EIDORS library approximates the non-linear DEIT forward problem (Equation 1) $F\left(\Delta a\right)=F\left(a\right)-F\left(a_{0}\right)$ between F(a) for the patient’s and $F(a_{0})$ for boneless thigh model by the transfer function F(Δa). This function represents a linear mapping [20] between ΔV, *I*, and Δa. The Jacobian matrix J describes the sensitivity of the individual measurements $\Delta V=V_{1}-V_{0}$ to changes Δa in the conductivity distribution, the current patterns *I*, the connectivity C, and the system matrices S(r) [47].

The norms ${∥∥}_{W}^{2}$ and ${∥∥}_{Q}^{2}$ represent the weighted L-2 norms of the residual error ΔV−F(Δa), and the conductivity difference Δa is used as regularization term. Typically, W is either represented by the identity matrix or the inverse covariance matrix of the measurements to model the knowledge about their reliability.

The inverse problem is, in general, a non-linear rank-deficient ill-posed problem. The term $∥a-a_{0}∥$ is used to regularize the solution by introducing a priori knowledge about the final conductivity distribution difference $\Delta a=a-a_{0}$. This knowledge is represented by regularization matrix Q. In the presented study, the Laplacian operator Q=Δ is used to achieve a distribution of Δa, which is smooth throughout the volume apart from the conductivity changes at the boundaries of the different tissues. The regularization parameter λ controls how much this constraint affects the obtained solution [48] and thus how smooth the final Δa will be. The residual error ε in (Equation 2) is minimized applying conjugated gradient minimization function provided by the EIDORS library. This function computes the minimum of ε using an approach along [49].

### 2.2. Segmentation of Bone Structures

In contrast to computed X-ray tomography or magnet resonance tomography, the electrical field generated by the injected currents is not strictly confined to the plane defined by the electrodes. It extends beyond the electrode plane upwards and downwards along a spherical path. Therefore, the measured voltages and thus the reconstructed conductivity difference images Δa also contain information about tissues located off the electrode plane. This information is essential for properly identifying the direction and orientation of the femur axis.

The standard approach in EIT is to generate an intermediate 2D slice image and use well-established 2D segmentation tools to extract the axis position. This works reasonably well within the electrode plane [39,40]. For estimating the orientation of the bone axis, it is necessary to reconstruct at least one or two slices off the electrode plane. Already, the decision at which distance these slices should be located depends upon various parameters, like the diameter of the thigh, its actual shape, and tissue composition, which influence how the electrical field is distributed within the thigh. The alternative is to directly segment the femur and extract its axis location and orientation from the 3D distribution of Δa within the thigh volume. In the presented computational study, a basic approach to directly extract the femur axis from the 3D distribution of Δa was chosen.

In the ideal case, the minimized conductivity difference Δa is zero throughout the whole thigh volume exempt for the elements representing the cortical bone and bone marrow. Depending on the tissues included in the model used to compute the estimate for $V_{0}$, elements collocated to the skin, the subcutaneous and intra-muscular fat, may also yield a Δa>0. Further noise in $V_{1}$, $V_{0}$, *I* and numerical inaccuracies in *J* may be amplified in the solution of (Equation 2), causing Δa≫0.

Therefore, the individual tetrahedral elements $\hat{▵}$ of the FEM are assigned to a bin $b_{i}$ out of 15 bins according to their final conductivity difference Δa value. The following approach is used to compute the corresponding bin boundaries based upon a sorted set $\Delta a_{sort}$ of the estimated conductivity differences Δa.

Remove any values of elements close to electrodes;Sort remaining Δa in ascending order and label sorted $\Delta a_{sort}$;Compute slope $\Delta a_{sl}=\Delta a_{sort,n+100}-\Delta a_{sort,n}$ for all *N* sorted samples;Pick all $\Delta a_{flat}=\left\{\Delta a_{sort}\;|\;\Delta a_{sl,sort}<\left(\Delta a_{sl,0}+\Delta a_{sl,N-100}\right)\frac{0.002N_{0}}{2N}\right\}$ including any $\Delta a_{sort}$ in between first and last $\Delta a_{flat}$;Fit line $y\left(x\right)=y_{0}+\left(x-x0\right)dy$ to $\Delta a_{flat}$ with $\left(x_{0},y_{0}\right)=median\left(\Delta a_{flat}\right)$;Compute $\Delta a_{rms}=\sqrt{{\left(\Delta a_{flat}-y\right)}^{2}}$Split the half-open interval $[\Delta a_{min}+\Delta a_{rms},y_{0}-\Delta a_{rms}]$ into $B_{lo}=9$ bins;Split the half-open interval $[y_{0}+\Delta a_{rms},\Delta a_{max}-\Delta a_{rms}]$ into $B_{hi}=5$ bins.

The number of bins $B_{lo}$ for the values below and $B_{hi}$ above the median section, the reference number of samples $N_{0}$ = 230,000, and all constant factors have been chosen by trial and error. The values $\Delta a=\Delta a_{flat}$ in bin $b_{10}$ correspond to tetrahedral elements $\hat{▵}$ for which the conductivity *a* does not differ from the value $a_{0}$ assumed in the forward model used to compute $V_{0}$. The corresponding conductivity difference Δa can be considered to be 0. The remaining bins $b_{1}$ to $b_{9}$ and $b_{11}$ to $b_{15}$ contain $\hat{▵}$ representing tissue parts for which the conductivity difference Δa between initial state conductivity $a_{0}$ and actual conductivity *a* is not zero.

These include transitional $\hat{▵}$ where Δa does not indicate a significant change in *a*. A proper assignment of these $\hat{▵}$ to either background representing muscle or any other tissue included in the boneless model or foreground is not possible; therefore, the following approach is used to properly separate foreground and background, discarding any $\hat{▵}$ which can not be reliably assigned.

Set the highest considered bin $b_{max}=7$;Select the first tetrahedral element ${\hat{▵}}_{seed}={\hat{▵}}_{i=0}$If all ${\hat{▵}}_{i}$ are either assigned or discarded, stop;If $b\left({\hat{▵}}_{seed}\right)>b_{max}$, discard ${\hat{▵}}_{seed}$ and continue from step 3 with ${\hat{▵}}_{seed}={\hat{▵}}_{i=i+1}$;If ${\hat{▵}}_{seed}$ has been assigned to a cluster *K*, continue from step 3 with ${\hat{▵}}_{seed}={\hat{▵}}_{i=i+1}$Create new cluster $K_{k=k+1}$;Assign ${\hat{▵}}_{seed}$ to new cluster $K_{k}$Collect all tetrahedral elements ${\hat{▵}}_{f=1-4}$ adjacent to the four faces of ${\hat{▵}}_{seed}$Discard any ${\hat{▵}}_{f=1-4}$ for which $b({\hat{▵}}_{f=1-4}>b_{m}ax$ holds;Skip any ${\hat{▵}}_{f=1-4}$ already assigned to a cluster *K*;Push remaining ${\hat{▵}}_{f=1-4}$ to work queue $q_{W}$ for further processing;If $q_{W}$ is empty, store cluster $K_{k}$ and continue from step 3 with ${\hat{▵}}_{seed}={\hat{▵}}_{i=i+1}$;Pop first ${\hat{▵}}_{q}$ from $q_{W}$ and continue from step 7 with ${\hat{▵}}_{seed}={\hat{▵}}_{q}$.

The conductivities $a_{bone}$ of cortical bone and bone marrow $a_{marrow}$ are the lowest of the modelled tissues. Therefore, it is assumed that all tetrahedral elements $\hat{▵}$ representing the femur bone are assigned to bins below bin $b_{10}$. Therefore, bins above $b_{10}$ are not considered at all in the selected approach. Through trial and error, it was found that bin $b_{9}$ contains transient elements with Δa>0 even though their actual *a* remains unchanged compared to the value assumed in $a_{0}$. Bin $b_{8}$ is discarded to form a distinct gap between background elements and those encoding bone and bone marrow starting at bin $b_{7}$.

The identified clusters $K_{k}$ of $\hat{▵}$ are sorted in ascending order according to the number of contained $\hat{▵}$. All clusters $K_{k}$ that contain less than 64 elements and are adjacent to the outer surface of the FEM model or to any of the electrodes are discarded.

### 2.3. Bone Position and Axial Orientation

In the final step, the central position and axial orientation is estimated for each of the remaining clusters, which are processed in ascending order of their number of contained elements.

Any $\hat{▵}$ for which the following condition with respect to mean $\bar{\Delta a}$ and standard deviation σ(Δa) of the conductivity differences Δa holds is discarded from the cluster $K_{k}$
(3)$$\bar{\Delta a}-1.9\sigma \left(\Delta a\right)<\Delta a<\bar{\Delta a}+1.9\sigma \left(\Delta a\right)$$

The remaining element centre points $p_{c}$ are used to compute an initial estimate for the central position $p_{b}$ and orientation $n_{b}$ of the bone.
(4)$$\begin{array}{c} U\Sigma V^{T}=svd\left(p-p_{b}\right) \end{array}$$
(5)$$\begin{array}{c} n_{b}=v_{1}^{T}\;p_{b}=mean\left(p\right) \end{array}$$
The estimate for $n_{b}$ is obtained by computing the singular value decomposition of all points $p_{c}$ It is assumed that the cluster corresponding to the bone has its largest extension along the bone axis $n_{b}$. Therefore, the first right eigenvector $v_{1}^{T}$ corresponding to the largest eigenvalue $\sigma_{1}$ is used as an initial estimate for n.

In a final step, a cylinder with radius $r_{0}=1.5\;cm$ is fitted to the cluster points minimizing the following function with respect to the centre *o* of the cylinder and its axis n
(6)$$\varepsilon_{r}=\frac{\sum (w^{2}{∥p_{c}-p_{b}-\zeta n_{b}∥}^{2})}{r_{0}^{2}}=min\;\mathrm{with}:\;\zeta =\langle p_{c}-p_{b}|n_{b}\rangle$$
The weights *w* are used to reduce the impact of specific points upon the final position $b_{b}$ and orientation $n_{b}$ of the fitted cylinder. This cylinder is shown in Figure 1b in yellow and its axis representing the estimated bone axis is symbolized as red line (Figure 1).

In the present study, all electrodes are mounted at the same height $z_{m}$ relative to the thigh model, forming a single electrode plane. Thus, the conductivity difference values Δa are less reliable for points located at a distance ζ far off the electrode plane [48]. A parabola parallel to the z-axis with its apex centred at the electrode plane $z_{m}$ is used to modulate the weights *w* accordingly.
(7)$$\begin{array}{c} f=\frac{\min{\left(z_{m}-z_{min},z_{max}-z_{m}\right)}^{2}}{4} \end{array}$$
(8)$$\begin{array}{c} h=\frac{1}{2}\left(\frac{{\left(z_{m}-\min\left(z_{el}\right)\right)}^{2}}{4f}+\frac{{\left(\max\left(z_{el}\right)-z_{m}\right)}^{2}}{4f}\right) \end{array}$$
(9)$$\begin{array}{c} w=\frac{{\left(\zeta -mean\left(\zeta \right)\right)}^{2}}{4f}+h \end{array}$$
The parabola spans from the minimum *z* coordinate $z_{min}$ to the maximum $z_{m}ax$ of all vertices in the model used to reconstruct Δa. The vertical shift *h* is the mean of the values obtained for the maximum $\max(z_{el})$ and minimum $\min(z_{el})$ vertices used to model the electrodes.

The function $\varepsilon_{r}$ is minimized by calling the SciPy [50] minimize function with the method parameter set to the L-BFGS-B [51] approach. Dependent upon the tissues included in the model used to compute $V_{0}$ and to compute the Jacobian matrix *J*, additional clusters *K* may be generated by the approach described in Section 2.2, e.g., for the intra-muscular fat. These typically have a smaller longitudinal extension $\Delta \zeta =\zeta_{max}-\zeta_{min}$ compared to the cluster $K_{b}$ corresponding to femur, which extends throughout the whole thigh. Therefore, the cluster *K* with the largest Δζ along $n_{b}$ is assumed to represent the femur. In case an additional cluster *K* with similar large Δζ exists, the *K* is selected which exposes lowest average mean conductivity difference $\bar{\Delta a}$ or covers the largest range ($\Delta a_{min},\Delta a_{max}$).

## 3. Results

In this study, five different thigh models were tested. They are either based upon a simple cylinder (Figure 2a) or a complex shape mimicking a more realistic cross section of the thigh. Figure 2b depicts the cross sections of the two computational FEM models of the thigh used to evaluate the suitability of the chosen DEIT-based approach to compute the position and orientation of the central femur axis. The simplistic cylinder models are used to validate implementation of the overall approach, verify the chosen system and reconstruction parameters, and test the direct 3D segmentation algorithm used to estimate the location and orientation of its central axis. They provided some basic findings and helped to gain some knowledge for interpreting the results obtained from the models mimicking a more realistic shape of the femur. An important question to be answered in this context is whether EIT-based methods can be used to locate the central femur axis and compute sufficiently distinct estimates regarding its position and orientation, symbolized in Figure 1 with the red and green lines.

For each of the two model types, a reference model was created for simulating the measured voltages $V_{1}$ by solving the DEIT forward problem (Equation 1).

All models included the conductivity $a_{skin}$ for the skin, the subcutaneous and intramuscular fat $a_{fat}$, the cortical bone $a_{bone}$, and bone marrow $a_{marrow}$. The values *a* chosen for each of the modelled compartments are shown in Table 1. They were selected according to [52].

For all simulations and measurements, an excitation frequency of 100 kHz was used. At this frequency, the conductivity of the modelled tissues are sufficiently distinct to obtain reasonable results for excitation currents I=10mA. The number of electrodes, selected for the simulations and dummy measurements as well as all other parameters like the regularization parameter λ were chosen in simulation experiments conducted prior to the presented computational study. Table 2 shows the parameter combination used to perform all the simulations studies and a first measurement using a test object cast from saline gel.

The stimulation current of 10 mA was chosen according to the electrical safety considerations at a stimulation frequency of about 100 kHz defined by the IEC 60601-1-1:2015 standard [53]:(10)$$I=\left\{\begin{array}{cc} 100\;\mu A_{rms} & 0.1\;\mathrm{Hz}<f<1\;\mathrm{kHz}\\ \frac{100\;\mu A_{rms}\;f}{1\;\mathrm{kHz}} & 1\;\mathrm{kHz}<f<100\;\mathrm{kHz}\\ 10\;mA_{\mathrm{rms}} & f>100\;\mathrm{kHz} \end{array}\right.$$

Each of the five thigh models tested correspond to one out of three possible configurations used to compute the reference voltages $V_{0}$ and to solve the inverse problem of DEIT (Equation 2). For testing and verification of the selected approach and to obtain some basic references, two models were based upon the cylinder model (Figure 2a). Three more realistic shaped (Figure 2b models were used for evaluating the feasibility of EIT-based location of bone axis with respect to its position and orientation.

The basic configuration 1 just represents a homogeneous conductivity distribution with $a=A_{muscle}=0.37\;\left.s/m\right.$ throughout the whole volume. Two setups (cylinder and realistic) along configuration 2 included the skin tissue and the subcutaneous fat. Configuration 3, which was used for the realistic model only, additionally included the intra-muscular fat.

The obtained positional deviations $\Delta p_{b}$ for each model and configuration are shown in Table 3. The estimates for the cylinder-based reference models deviate by ≈1.4–2.8 mm from the actual bone position. By fitting a cylinder, $\Delta p_{b}$ could be reduced to $\Delta p_{b}\approx 1\;mm$. For the realistic model, the initial position estimates are off by 7.9 mm in the homogeneous case (configuration 1) and by 24.8 mm for configuration 2. Fitting a cylinder, as symbolized by the yellow cylinder surface in Figure 1b, does not further reduce $\Delta p_{b}$ in contrast to the cylinder models. On average, a deviation of ≈1 cm was achieved between the true and estimated positions. Table 3 also shows the angular deviation φ between the actual femur axis $n_{f}$ and the reconstructed axis $n_{b}$. On average, the estimated $n_{b}$ deviates by φ=1.50°.

## 4. Discussion

The results obtained in this computational simulation study are quite promising. Especially the small angular deviations of φ≤2.9° clearly indicate that the DEIT approach is suitable for extracting the position and axial orientation of the femur and tibial bones inside a patients lower limb. Similar results regarding the axial position were found by [39,40] using electrical capacitance and resistance tomographic approach. They used cross section slices of their simple cylindrical thigh model consisting of simulated cortical femur bone and muscle as well as their ex vivo model consisting of the upper part of a cemented femur bone and saline gel to locate its axis. With this approach, they achieved absolute positional errors between 0.2 mm and 1.3 mm, similar to the ones obtained in the presented study for the cylinder models. In [39,40], the orientation of the femur shaft is assumed to be parallel to the z axis of the electrode plane in any case and thus no values for the angular deviation of its axis are provided.

Compared to the small deviations of ≈1 mm for the cylinder models, rather large positional deviations of up to 2 cm were encountered in the presented study on the realistic models. In the following, some of the aspects causing this huge differences and possible solutions to avoid them in the envisioned application are discussed. The shape of a realistic cross section of a patient’s thigh was generated by deforming he the cylinder model, thereby the overall shape and the shape of the inner cylinder modelling the bone were distorted the same amount. The resulting change in diameter and shape of the simulated bone likely adds to the larger radial deviation Δo between its true centre and the centre used to compute the positional deviation Δo between the reconstructed $o^{\prime }$ and actual position *o*. Part of this effect already was mitigated in this study by selecting the centre $o_{m}$ of the bone marrow as the reference point instead of the overall bone *o*.

Another limitation is caused by the fact that, in the present study, a standard DEIT approach is used to model the forward and inverse inverse problem (Equation 2). Both are modelled as linear mappings between the conductivity difference $\Delta a=a_{1}-a_{0}$ and the difference $\Delta V=V_{1}-V_{o}$ of the measured voltages for actual distribution *a* and boneless state $a_{0}$. According to [54] this linearization is only applicable if Δa is small with respect to $a_{0}$. From Table 1 it easily can be deduced that the conductivity contrasts $\left.|a_{bone}-a_{muscle}|/a_{muscle}\right.=95\%\gg 20\%$ between muscle $a_{muscle}=0.37\left.S/m\right.$ and cortical bone $a_{marrow}=0.02\left.S/m\right.$$ and $\left.|(a_{marrow}-a_{muscle}|/a_{muscle}\right.=99\%\gg 20\%$ between bone marrow $a_{marrow}=0.002\left.S/m\right.$, respectively, are clearly dominated by $a_{muscle}$. This clearly indicates that linearization is not applicable to the inverse problem (Equation 2) used to find the position and orientation of femur and tibia inside a patients thigh or lower leg. Therefore, the achieved positional deviations of ≈1 mm for the cylinder models and 5–25 mm for the realistic models are very encouraging.

Given the large conductivity contrasts of ≫20%, computing sufficiently precise estimates of the absolute conductivity values within a patients thigh is essential for the envisioned application. This can be achieved using an absolute EIT algorithm. Up to date, none of these algorithms performed sufficiently well within clinical applications [19]. An approach which allows to compute the absolute conductivity values $a_{abs}$ while offering a sufficiently robust performance in clinical applications will lead to a further reduction in the positional and angular deviations. In the envisioned application, the state $s_{0}$ without bone can not be directly assessed through direct measurements and thus has to be simulated using an according model. The corresponding voltage distributions $V_{0}$ are computed by assuming the corresponding conductivity distribution and solving the EIT forward problem (Equation 1). Given the assumption that the admittance Y(a) can be split into a term $Y_{const}$ which does not depend upon *a* and the term X=1/D(a), containing only the matrix D(a), the inverse problem of DEIT (Equation 2) can be expressed as follows: (11)$$\begin{array}{c} \varepsilon ={∥\Delta V-\Delta V=\frac{I}{Y_{const}}X∥}^{2}+\lambda^{2}{∥a-a_{0}∥}^{2}=min \end{array}$$(12)$$\begin{array}{c} X=\left(\frac{1}{D(a)}-\frac{1}{D(a_{0})}\right) \end{array}$$
As $a_{0}$ is known, it should be possible to compute absolute values for $a_{abs}$ after directly solving (Equation 11) The obtained $a_{abs}$ then would allow to estimate the position and orientation of the femur and as well as the tibia with deviations of only a few millimetres in position and less than 2° angular deviation from its true orientation.

The first verification tests using saline gel casts did not yield any proper results. The root cause was that some of the electrodes had no proper electrical contact to the surface of the saline gel dummy. This went unnoticed until the recorded data were finally analysed and reconstruction of the position of the hole simulating the femur was attempted. Therefore, alternative approaches to manufacture testing dummies of the thigh and lower leg are currently evaluated. For example, they could be 3D printed, as suggested in [55].

## 5. Conclusions and Outlook

The initial hypothesis of the present study was that including a priori knowledge about the different types of tissues and their location within the thigh improves the accuracy of the estimated bone position and direction. According to Table 3, the smallest positional deviations Δ0 are obtained for the homogeneous configuration 1. For the realistic models, the positional deviations are about half the size or less than those of configurations 2 and 3 where the conductivity distribution $a_{0}$ includes the skin and fat. These results clearly indicate that any anatomical a priori knowledge should be used to compute non-zero initial values $\Delta a_{0}$ for Δa instead of including them in the model used to simulate $V_{0}$. For computing $V_{0}$ to compare the measured data $V_{1}$ the homogeneous models (configuration 1) should be sufficient.

This is, apart from the rather disappointing outcomes of the gel dummy measurements, quite promising and encouraging. However, the clinical application, envisioned in Figure 1, is highly sensitive to variations in the axis position and orientation. Even small placement errors caused by improper axis estimates can result in non-desired alterations [56] of the knee joint mechanics established by the prosthetic components. Therefore, further efforts are necessary to achieve positional and angular precision’s of less than a few mm and <2°.

This includes the presented approach to directly segment the distribution difference Δa in 3D allowed to compute reasonable estimates of the femur axis orientation. Its advantage compared to the standard cross section slice approach is [33,35,40,41] that it directly assess the related information encoded in the electrical field expanding beyond the electrode plane. A more elaborated 3D approach will be necessary to achieve the accuracy of the positional and axial orientation required for the envisioned application and to avoid any related undesired alteration in the restored knee joint mechanics [56].

Currently, further studies and tests including artificial test bodies, ex vivo experiments, and test on humans are prepared. The goal is to increase the achieved position and angular accuracies and to evaluate more advanced approaches to directly solve the non-linear inverse problem of DEIT.

It will be tested whether the non-linear DEIT approach, roughly outlined and briefly discussed in Section 4, is suitable for computing sufficiently precise absolute values of the conductivity distribution *a* in the patient’s thigh and at the same time whether it shares the robustness and stability with respect to clinical application with well-established DEIT algorithms [19].

In this context, it will be tested whether and how non-zero initial estimates for $x_{0}$ representing the initial differences between boneless state conductivity distribution $a_{0}$ and the actual *a* affect the radial and angular deviation of the bone position and orientation. A main focus will be on how a priori anatomical information about the tissues present in the limb, can be modelled based upon the individual tissue conductivities $a_{tissue}$.

## Figures and Tables

**Figure 1 sensors-24-05269-f001:**
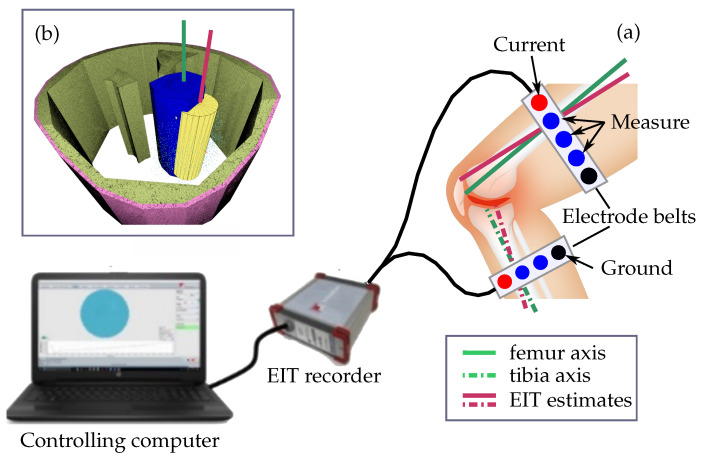
The electrical impedance tomography system used to find the position and orientation of the axis of the femur axis consists of (**a**) the electrode belts, the EIT recorder, and a controlling computer. The EIT recorder generates the measurement currents applied through the red electrodes and measures the voltage differences at the remaining electrodes resulting from the corresponding electrical field. The black electrodes symbolize the common ground. A standard differential EIT approach is used to estimate the conductivity difference between a simulated reference measurement form a thigh without and the measured data. The resulting 3D relative conductivity distribution map is segmented to find the finite elements representing the femur. The estimate for the position and orientation of its axis (**b**) is computed by fitting a cylinder (yellow). The accuracy of the obtained results is evaluated by computing the positional offset and the angular deviation between the estimated (red line) and the actual position and orientation of the femur axis (green line). The envisioned assessment of the tibia axis is indicated by the dashed and dotted lines.

**Figure 2 sensors-24-05269-f002:**
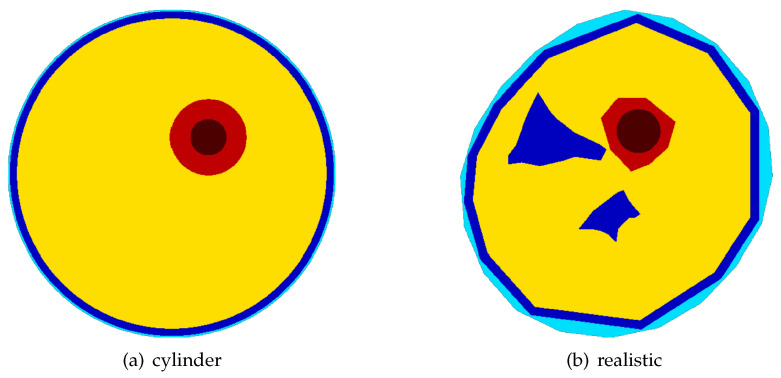
The two types of thigh models used: a simple cylindrical model (**a**) and a complex shape (**b**) mimicking a realistic cross section of the thigh. The cylinder model is used to validate the chosen approach, test the algorithm used to locate the bone axis, and verify the obtained results. The most complex models included the skin (light blue), fat (blue), cortical bone (red), bone marrow (dark red), and muscle (yellow).

**Table 1 sensors-24-05269-t001:** Conductivity values for the tissues considered in the present study. The values were selected for an excitation frequency 100 kHz according to [52].

Tissue	Skin	Fat	Muscle	Bone	Marrow
Conductivity $\left[S/m\right]$	0.065	0.03	0.37	0.02	0.002

**Table 2 sensors-24-05269-t002:** The EIT system configuration, simulation, and reconstruction parameters used throughout this study.

Electrodes	Number	16
	Type	Ag/AgCl
	Diameter	7 mm
Stimulation	Pattern	opposite electrodes
	Current	10 mA
Regularization	λ	0.00007

**Table 3 sensors-24-05269-t003:** The positional and axial deviations obtained for the different thigh model configurations. Two configurations were tested for the cylinder model and three for the model mimicking a realistic thigh shape. The third configuration tested for the realistic shaped model includes the intra-muscular fat in addition to the subcutaneous fat compartments.

	Tissue Configuration				$\Delta p_{b}\left[\mathrm{mm}\right]$		φ[°]	
Model	Muscle	Skin	Fat		Initial	Optimal	Initial	Optimal
			Subcut.	Inter				
Cylinder	X	–	–	–	1.4	1.1	0.72	1.14
	X	X	X	–	2.8	1.3	0.86	0.20
Realistic	X	–	–	–	7.9	8.0	5.42	2.90
	X	X	X	–	24.8	21.0	1.76	0.56
	X	X	X	X	14.0	14.4	3.39	2.63
mean	–	–	–	–	10.2	9.2	2.43	1.49

## Data Availability

The data and software are made available in the data and software repository of Johannes Kepler university. Until public access to these reporsitories is available in its full extent data developed data and software are provided on request by the institute for Biomedical Mechatronics at Johannes Kepler University, mmt@jku.at

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
