# Peer review of "DEIT-Based Bone Position and Orientation Estimation for Robotic Support in Total Knee Arthroplasty—A Computational Feasibility Study"

_sensors, 2024, doi:10.3390/s24165269_

Round 1

Reviewer 1 Report

Comments and Suggestions for Authors

In this study time differential electrical  impedance tomography is used to locate the femur positions. The overall  approach was tested by simulating  different configurations of thigh shape and considered the different tissue  conductivities. For the different models the reconstructed position/angle deviations   from the actual bone center was calculated.

The paper is interesting and suitable for publication.

However some improvements are necessary:

1)      The observed deviations from the real values are important  or not? Are they detrimental for the foreseen application or not?

2)      A comparison between the evaluation obtained with this system and the others have to be added in a specific paragraph

Reviewer 2 Report

Comments and Suggestions for Authors
  1. What is the meaning and purpose of Figure 1?  This is EIT in the knee and the picture from another reference on the liver.
  2. Section 2 and section 2.1 are very basic out-of-shelf methods provided by the EIDORS software. No new and valuable approach is shown there. 
  3. The segmentation task then led to a very simple model in Figure 3.
  4. Justification of 10 mA current was not given.
  5. The conclusion in Table 3 is trivial. No one will use a cylinder model for geometry like the knee.  This would have been the case about 30 years ago.
  6. In literature, there is some good work using EIT for imaging bones and they have been done in real people.  So this pure simulation study has 0 value.
  7. A lot of experiments are shown in open hardware EIT: EIT-kit: An Electrical Impedance Tomography Toolkit for Health and Motion Sensing (mit.edu) MuscleRehab: Improving Unsupervised Physical Rehabilitation by Monitoring and Visualizing Muscle Engagement (mit.edu)
  8. And there are many more.

Reviewer 3 Report

Comments and Suggestions for Authors

In this simulation study, differential electrical impedance tomography (DEIT) is used to estimate bone position and orientation in different computational thigh models. Further developments in this area may lead to applications for robotic support in total knee arthroplasty.

The work involves mainly computational modelling utilizing anatomical data. "Discussion" and "Results" sections focus on various aspects of the simulation.

The concept still needs to be developed significantly with physiological and clinical inputs before relating it directly to its surgical applications. 

Therefore,

- the title of this work may include "Computer" or "Computational" to clarify on the kind of "Simulation Study". This is expected to give the reader a more realistic context of the work.

- it is better to mention in the discussion how the tissue impedance is influenced in a living person, particularly related to the bone applications.

- it is also better to include some clinical aspects. For example, studies have shown that small errors in placement of prosthetic components can alter the knee joint mechanics. (https://doi.org/10.1142/S0219519412400027).

- English language needs attention. For example (these are only few examples). "atrhroplasty" in the title; "educe" on page 3, line 60; "of the subjects thigh" page 4, line 83.

- does the symbol 'a' refer to conductivity distribution or potential distribution (line 78, page 4)?

- There is no obvious section on "Conclusions".

- Though the technical work is well described and has good conceptual value, the submitted draft could have been much better. 

Comments on the Quality of English Language

Needs revision.

Round 2

Reviewer 2 Report

Comments and Suggestions for Authors

Some good improvement is made on this version.

One remaining question:

It is known EIT works well for time-difference imaging.

If I remember correctly, none of the EIT's medical applications worked by absolute imaging. 

Here we need absolute-value imaging, and it is known to be possible in computational feasibility.

Can you please show how these could be done in experimental studies and more critically in clinical applications?

Reviewer 3 Report

Comments and Suggestions for Authors

The manuscript is now much improved except for spelling mistakes and formatting.

Comments on the Quality of English Language

Quality of English language is OK. However, spellcheck is required.
